# Does Artificial Intelligence Make Clinical Decision Better? A Review of Artificial Intelligence and Machine Learning in Acute Kidney Injury Prediction

**DOI:** 10.3390/healthcare9121662

**Published:** 2021-11-30

**Authors:** Tao Han Lee, Jia-Jin Chen, Chi-Tung Cheng, Chih-Hsiang Chang

**Affiliations:** 1Kidney Research Center, Department of Nephrology, Chang Gung Memorial Hospital, Linkou Branch, Taoyuan 33305, Taiwan; kate0327@hotmail.com (T.H.L.); Raymond110234@hotmail.com (J.-J.C.); 2Graduate Institute of Clinical Medical Science, College of Medicine, Chang Gung University, Taoyuan 33302, Taiwan; 3Department of Trauma and Emergency Surgery, Chang Gung Memorial Hospital, Taoyuan 33305, Taiwan; atong89130@gmail.com

**Keywords:** artificial intelligence, machine learning, acute kidney injury, prediction model

## Abstract

Acute kidney injury (AKI) is a common complication of hospitalization that greatly and negatively affects the short-term and long-term outcomes of patients. Current guidelines use serum creatinine level and urine output rate for defining AKI and as the staging criteria of AKI. However, because they are not sensitive or specific markers of AKI, clinicians find it difficult to predict the occurrence of AKI and prescribe timely treatment. Advances in computing technology have led to the recent use of machine learning and artificial intelligence in AKI prediction, recent research reported that by using electronic health records (EHR) the AKI prediction via machine-learning models can reach AUROC over 0.80, in some studies even reach 0.93. Our review begins with the background and history of the definition of AKI, and the evolution of AKI risk factors and prediction models is also appraised. Then, we summarize the current evidence regarding the application of e-alert systems and machine-learning models in AKI prediction.

## 1. Introduction

Acute kidney injury (AKI), defined as increased serum creatinine level or decreased urine output, is the most common and adverse complication of hospitalization in patients [1]. The incidence of AKI among inpatients ranges from 5% to 10%, and it ranges from 20% to 70% among patients admitted to an intensive care unit (ICU) [2,3,4,5]. AKI incidence varies by clinical condition; approximately 20% of patients with Stevens–Johnson syndrome or toxic epidermal necrolysis developed AKI, and 56% of patients with severe sepsis developed AKI. Among patients who have undergone surgery, AKI incidence varies by the type of operation, ranging from 25% for trauma surgery to as high as 50% for cardiac or aortic surgery [6,7]. Although the quality of medication data and the effectiveness of treatment have greatly improved recently, the incidence of AKI has continually increased, possibly due to the aging population and rising comorbidities, such as diabetes mellitus and hypertension.

After an initial AKI episode, the risk of chronic kidney disease (CKD), long-term dialysis and mortality are significantly increased in the affected patients [8,9,10,11,12,13,14]. According to a previous meta-analysis, patients with AKI had higher risks of CKD, end-stage renal disease (ESRD), and mortality than patients without AKI; the hazard ratios were 8.8, 3.1, and 2.0, respectively [10]. Among patients with AKI, those with dialysis-dependent AKI had even poorer renal outcomes than patients with non-dialysis-dependent AKI [14,15]. Although investigators had identified that patients with hypertension or diabetes mellitus, those requiring readmission for cardiovascular disease or sepsis, those receiving cardiovascular surgery or neurosurgery, and those taking nephrotoxic agents (nonsteroidal anti-inflammatory drugs, radiocontrast, hydroxyethyl starch, and nephrotoxic antimicrobials) were prone to experience AKI [16,17,18]. No accurate tool has been established for identifying patients at risk of AKI and for predicting AKI occurrence. At the same time, patients only exhibit imperceptible signs of AKI or even exhibit no clinical symptoms in the early stages of AKI. Once oliguria, hematuria, or anasarca is present, patients may already have considerable parenchymal injury and require renal replacement therapy. Although research on novel biomarkers has increased in recent years, advances in clinical informatics, artificial intelligence (AI), and machine learning may enable the development of additional approaches for the prediction and estimation of AKI risk through the processing of electronic medical records (EMRs) [19]. In this article, we review the progress in the application of machine learning systems for AKI risk prediction.

### 1.1. AKI Definition

The definition of AKI has evolved over the past few decades, ranging from the initial Risk, Injury, Failure, Loss of kidney function, and End-stage kidney disease (RIFLE) classification and the Acute Kidney Injury Network (AKIN) criteria to the most recent Kidney Disease Improving Global Outcome (KDIGO) guidelines [1,20,21]. The KDIGO guidelines have been the most widely used definition of AKI over the past decade, and according to these guidelines, AKI is divided into stages by severity on the basis of increasing serum creatinine level and urine output rate data. However, the serum creatinine level and urine output rate are not sensitive or specific markers of AKI. The interpretation of changes in renal function is prone to error when conducted on the basis of serum creatinine level. First, because creatinine is not only glomerular-filtered but also secreted by tubules, creatinine clearance overestimates the true GFR, especially in cases of decreased renal function [22,23]. Second, serum creatinine level is influenced by muscle mass (creatinine is a product of muscle catabolism), diet (a protein-rich diet results in higher serum creatinine level), and drugs (for example, trimethoprim and cimetidine interfere with the tubular secretion of creatinine) [24,25]. Third, the production of muscular creatine is influenced by disease status; for example, it is lower and greater in severe hepatic disease and rhabdomyolysis, respectively [22,26]. Lastly, serum creatinine level is not significantly elevated until 48 h after renal injury, and delayed elevation detrimentally affects the timely identification of renal injury [27,28]. Although urine output rate may reflect renal function decline in a timelier manner, it is still affected by the patient’s volemic status and is influenced by diuretic treatment.

Because both serum creatinine level and urine output rate are nonspecific and inaccurate markers of AKI, multiple novel biomarkers have been investigated for predicting or diagnosing AKI in a timely manner. The following novel biomarkers have been identified for the early detection of AKI: cystatin C, neutrophil gelatinase-associated lipocalin, kidney injury molecule 1, liver type fatty-acid binding protein, urine angiotensinogen (AGT), and calprotectin. Chen and colleagues reported that serum cystatin C, urine NGAL, and serum interleukin-18 (IL-18) played valuable roles in the early detection of AKI in a cardiac care unit (CCU) and that the areas under the receiver operating characteristic curve (AUROCs) of serum cystatin C, urine NGAL, and serum IL-18 for AKI prediction were 0.895, 0.886, and 0.841, respectively. Multiple regression analysis indicated that urine NGAL, serum IL-18, and sodium levels at CCU admission were independent risk factors for 6-month mortality. Among these factors, urine NGAL had the highest discriminatory power, and the Youden index indicated that it yielded the most accurate prediction of patient mortality [29]. Some studies have described pseudo-worsening renal failure (also termed pseudo-AKI), which is a common clinical condition in patients with cardiorenal syndrome in which increases in serum creatinine level are induced by diuretic treatment rather than by tubular necrosis or interstitial nephritis. These studies have suggested that the novel biomarker calprotectin can distinguish a true AKI episode from a pseudoepisode of diuretics-related AKI [30,31]. Chang et al. aptly reported that calprotectin had an excellent AUROC of 0.946 for predicting intrinsic AKI [32].

Although the novel AKI biomarkers identified in recent studies have greatly improved and enabled the earlier detection of AKI, many difficulties remain in applying these biomarkers in clinical settings. Vanmassenhove and colleagues noted that the early diagnosis of AKI by using novel serum and urinary biomarkers remains cumbersome, especially in settings in which the timing and etiology of AKI are not well defined [33]. Another difficulty is that tests for novel biomarkers are not widely commercially available or can be expensive and repeat examinations may be required during the process of AKI diagnosis. Moreover, Marx et al. concluded that it is almost impossible to depend on one universal serum or urine biomarker to determine the risk, diagnosis, severity, and outcome of AKI and to discriminate between etiologies of AKI and monitor its course [34]. AKI is a nonuniform, complex condition with a wide spectrum of causes and pathophysiological mechanisms; therefore, the requirement of several biomarkers or marker panels that cover different aspects of AKI seems reasonable for standardizing diagnoses [34,35]. However, examining multiple novel biomarkers or evaluating the patient’s condition by using marker panels may further increase the costs of predicting or diagnosing AKI early and accurately. Therefore, the most cost-effective method appears to be identifying which patients with AKI are at high risk before arranging a biomarker examination for them.

### 1.2. AKI Risk Factors and Risk Scores

Some studies that have focused on identifying significant risk factors for AKI have determined that both patient susceptibilities and exposure are crucial in AKI development. Patient susceptibilities include age, gender, race, and comorbidities. Among all comorbidities, CKD has been identified as a major risk factor for AKI due to its associated loss of autoregulation, loss of renal reserve, and susceptibility to nephrotoxic agents. Moreover, diabetes mellitus, hypertension, cardiovascular disease, hyperuricemia, obesity, and liver disease have all been reported as risk factors for AKI [19,36,37]. Exposure to sepsis, nephrotoxic agents, surgical intervention, and shock have been identified as contributors to AKI [16,17]. A multicenter international cross-sectional AKI–EPI study reported that sepsis, hypovolemia, and nephrotoxic drug exposure were the three most frequently reported etiologies of AKI in patients with a critical illness [16]. The incidence of AKI may be higher among patients with poor physical condition after certain exposure; for example, an aging patient may have a higher risk of AKI after cardiac surgery. However, AKI risk differs by the physical condition and nephrotoxic exposure; this renders accurate risk assessment challenging.

After the risk factors for AKI were identified, investigators began focusing on establishing a risk score by using a combination of independent AKI predictors, assessment of relative impact, and external validation. A precise risk prediction score must be able to identify at-risk patients and guide physicians in preventing, diagnosing, and treating the disease. Different scoring systems have been constructed for assessing the risk of AKI in specific groups of patients; these prediction models include age, gender, baseline renal function, and comorbidities, and specific predictors can be added depending on surgery type, medication, and procedure-related data.

The Mehran risk score was proposed in 2004 for analyzing the risk of AKI and the requirement of renal replacement therapy in patients with postpercutaneous coronary intervention; according to later external validation conducted in 2016, the system exhibited adequate performance for predicting contrast-induced nephropathy in patients with acute coronary syndrome who underwent coronary angiography [38,39]. Large cohort studies have revealed that surgery is a major cause of AKI, and the AKI incidence rate ranges from 25% for trauma surgery to as high as 50% for cardiac or aortic surgery [6,7,40]. Additionally, cardiac surgery is associated with the highest AKI incidence among all types of surgery, ranging from 2% to 50%, and the dialysis-dependent rate is 1% to 6% [41,42]; therefore, it is unsurprising that several prediction models have been established for AKI risk identification in patients who plan to undergo cardiac surgery. The earliest scoring system EuroSCORE is based on European multicenter data published in 1999, and the 2010 Value of Age, Creatinine, and Ejection Fraction (ACEF) score is also based on data from European databases [43,44]. The short-term risk (Society of Thoracic Surgeons, STS) score was created in 2008 by using data from the national database of the American Society of Thoracic Surgery; this score is used to evaluate adult preoperative cardiac surgery risk, and professionals have retained and modified this prediction model [45,46]. In an externally validated study, 196 patients received mitral valve repair, and their STS and ACEF scores were compared; the STS renal failure score was the most accurate for predicting stage 2 and 3 AKI. Additionally, that study found that ACEF scores exhibited an AUROC similar to that of STS renal failure scores across all AKI predictions (ACEF and STS score AUROCs: 0.758 and 0.797, respectively), but the ACEF score includes only three prediction factors: age, creatinine, and ejection fraction; thus, the ACEF score is more convenient for clinical physicians [41]. In another study that compared the preoperative risk models of AKI in isolated coronary artery bypass grafting surgery, the EuroSCORE II, STS score, and ACEF score all performed adequately for predicting stage 3 AKI; additionally, the ACEF score exhibited satisfactory discriminatory power for predicting postoperative AKI, with an AUROC of 0.781 [47].

Besides the comorbidities and acute illness conditions, race and epidemiology factors also showed their impact on AKI incidence according to previous studies. Mathioudakis and his colleagues had reported that blacks had a 50% higher age- and sex-adjusted odds of AKI compared to whites (odds ratio: 1.51; 95% CI 1.37–1.66) based on the national databases of the U.S. This association between the black race and increased risk of AKI persisted after additional adjustment for multiple AKI-related risk factors [48]. In 2013, a meta-analysis focused on AKI incidence worldwide reported that the pooled rate of AKI according to KDIGO criteria showed a difference around the world. According to geographic regions of the world and patterns of country economies and latitude, the pooled rate of AKI appeared higher in South versus North America (29.6% versus 24.5%), Southern versus Northern Europe (31.5% versus 14.7%), and South versus Western or Eastern Asia (23.7% versus 16.7% versus 14.7%). The pooled rate of AKI appeared higher in studies from countries located south versus north of the equator (27.0% versus 22.6%), in addition, this study also revealed that the AKI incidence was high in countries that spent >10% versus ≤5% GDP on total health expenditure (25.2% versus 14.5%) [49]. 

Considering the influence of race and epidemiology on AKI incidence, some investigators have validated their scores against data from their country’s health insurance research database to achieve high prediction performance. An example is the ADVANCIS score, which is used to predict AKI in patients who receive percutaneous coronary intervention (PCI) for coronary artery disease; the score was validated against data from Taiwan’s National Health Insurance Research Database. The ADVANCIS score uses eight clinical parameters (age, diabetes mellitus, ventilator use, prior AKI, number of intervened vessels, CKD, IABP use, and cardiogenic shock), and the score ranges from 0 to 22; additionally, an ADVANCIS score of ≥6 is associated with higher in-hospital mortality risk [50]. In addition to modifying risk prediction models in accordance with epidemiological factors, researchers have included novel biomarkers as prediction factors in some modern AKI prediction score systems and have assessed the association between biomarkers and patients’ clinical information. Zhou et al. established a prediction score of AKI in patients with acute decompensated heart failure by setting urine NGAL and urine AGT as risk factors [51].

Although various scoring systems have been established to address different clinical conditions, most prediction models can perform only as single-point AKI prediction models, such as predicting AKI incidence after a specific type of surgery or before the use of a contrast agent, making it difficult to reflect changes in real-time. Furthermore, some of these scoring systems cover several factors, including baseline condition, clinical data, and novel biomarkers, making them too complex for clinical use. With the development of information technology, some hospitals have integrated these prediction systems into their medical informatics systems (MISs), and these clinical risk assessment tools have been increasingly used because they enable the automated analysis of data. Because race, genes, disease prevalence, and medication differ between countries, the combined use of an MIS and risk prediction scores potentially enable the use of data from local databases to assess the risks of AKI and the requirement of renal replacement therapy.

### 1.3. From Automated Electronic Alerts to AI

As MISs become more popular, systems that provide automated electronic alerts (e-alerts) have become increasingly feasible; in such a system, the electronic records and clinical information of patients are analyzed using an algorithm that predicts whether early or subclinical AKI is present [52]. These systems are expected to aid patient care by making clinical evaluation and treatment timelier. Park and his colleagues had investigated an AKI alert system with automated nephrologist consultation in which clinicians could generate automated consultations to the nephrology division while patient’s serum creatinine concentration elevation of at least 1.5-fold or 0.3 mg/dL from baseline. This study reported that the early consultation with a nephrologist was greater (adjusted OR, 6.13; 95% CI, 4.80–7.82) and odds of a severe AKI event were reduced (adjusted OR, 0.75; 95% CI, 0.64–0.89) after introducing the e-alert system. However, mortality was not affected (adjusted HR, 1.07; 95% CI, 0.68–1.68) [53]. Another study used an e-alert system in ICU patients, clinician received a “pop-up” message while the e-alert system screened the serum creatinine data and detected possible AKI events following the KDIGO criteria definition. Although the sensitivity, specificity, Youden Index and accuracy of the AKI e-alert system were 99.8, 97.7, 97.5 and 98.1%, respectively, in this study, and the prevalence of diagnosis AKI and the prevalence of nephrology consultation in the e-alert group was higher than that in the non-e-alert group. There was no significant difference in the prevalence of dialysis, rehabilitation of renal function, or death in the two groups [54]. In 2017, a systemic review concluded that an e-alerts system neither reduced mortality (odds ratio [OR], 1.05; 95% CI, 0.84–1.31) nor reduced the incidence of dialysis treatment (OR, 1.20; 95% CI, 0.91–1.57) [55,56]. All six studies included in this meta-analysis used only serum creatinine change as the trigger for e-alerts, and serum creatinine change is neither a sensitive nor specific marker of kidney injury, as mentioned in the preceding paragraph. Beyond the limitation of serum creatinine as an AKI marker, e-alerts systems face challenges when used in patients without baseline renal function and those with CKD who have higher baseline creatinine levels and more significant changes in renal function following small changes in creatinine level; a wide variety of further care is provided by clinicians to patients after the receipt of e-alerts. To prescribe standardized and evidence-based clinical care after the receipt of e-alerts, a care bundle was built. The most recent guidelines prescribe no specific management options for AKI, and the treatment strategy is mainly supportive. In critically ill patients, the occurrence and severity of AKI were reduced following adherence to KDIGO guidelines detailing the management of fluids, avoidance of nephrotoxins, monitoring of serum creatinine levels and hemodynamics, and referral to a specialist. Several studies have reported a decrease in hospital-acquired AKI and AKI-associated mortality and hospitalization days when the e-alert system was combined with a care bundle, the patient’s history was analyzed, the patient’s urine samples were tested, a clinical diagnosis of AKI was established, the course of treatment and testing was planned, and advice was sought from a nephrologist [57,58,59]. Machine-learning algorithms are in high demand and require large volumes of data. With large EMR databases and powerful computing hardware, scholars have extended the application of machine learning. Recently, AI has also been applied with various machine-learning algorithms, especially deep neural networks.

## 2. Methods

In order to get a closer look at the investigating of machine-learning studies on AKI prediction, we searched PubMed for clinical trials and conference abstracts discussing how machine learning and AI can be used to predict AKI. Online literature searches of the PubMed database were performed, and the database search was last updated on 1 December 2020. The search strategy targeted published clinical trials, including conference abstracts that described the use of machine learning for predicting AKI in adults. The search strategy and results are detailed in Appendix A. Two investigators (T.H. Lee and J.J. Chen) independently evaluated the titles and abstracts of the retrieved studies, and articles were excluded upon initial screening if their titles or abstracts indicated that they were clearly irrelevant to the objective of the current study. Full-text reviews were then performed for the articles deemed potentially relevant to assess their eligibility for inclusion. The study inclusion criteria were as follows: (i) a study population consisting of adults and the study having a prospective or retrospective design and (ii) AKI prediction through machine learning. Case series and reports, conference abstracts, comments on other studies, and review articles were excluded.

## 3. Results

In total, 31 studies reported the discriminating ability of machine learning for predicting AKI (Table 1).

As shown in Table 1, the included studies predicted AKI adequately, some studies had AUROC > 0.8, and the study conducted by Koola et al. had the highest AUROC of 0.93 in logistic regression. The models outperformed diagnosis through novel biomarkers. Machine-learning models that were used to predict AKI had four to 57 covariates. These covariates were epidemiological factors, comorbidities, laboratory data, medications, and surgery types. We summarize the most commonly used covariates in these machine-learning prediction models in Figure 1. In the 31 studies, the five most commonly used covariates were creatinine, age, blood pressure, gender, and diabetes mellitus. Among these 31 studies, eight studies focused on patients’ undergoing surgery (surgeries were cardiac or aortic surgeries in five studies), and the most commonly used covariates in surgical patients are illustrated in Figure 2; the five most common used covariates were gender, body mass index, age, creatinine, and surgery type.

In Table 2, we summarized the method of feature selection, data splitting and machine learning algorithm choices in enrolled studies. Different performances on predicting AKI by using different machine learning algorithms were also listed in this table. More than half of the enrolled studies used LASSO, XGBoost, or other feature selection methods to choose the covariates for machine learning, but some studies chose covariates according to clinical experience or previous reports.

## 4. Discussion

Among these 31 studies, there were several studies that are worth addressing. By reviewing these studies, we found that most of these studies lacked external validation, which implies that the results cannot be extended to other populations. Two studies performed external validation. Simonov and colleagues established a real-time AKI prediction model by using an electronic health record (EHR) dataset of 169,859 hospital admissions in three hospitals. The training dataset contained the data of 60,701 patients, and the internal validation dataset contained the data of 30,599 patients from the same hospitals; external validation was performed with the data sets of 43,534 and 35,025 patients from two other hospitals. The incidence of AKI was similar in the training and external validation datasets (19.1% and 18.9%, respectively). Discrete-time logistic regression was used to train the model, a total of 35 covariates were included in the fully adjusted models, and the AUROCs for predict sustained AKI, dialysis, and death were 0.77 (95% CI, 0.76–0.78), 0.79 (95% CI, 0.73–0.85), and 0.69 (95% CI, 0.67–0.72), respectively [69,91]. This real-time prediction model was based on large cohorts including patients requiring hospitalization and those in surgical and ICU settings, and the external validation of this model was performed using the data from two other institutions, with high predictive performance found across the three diverse care settings; the subsequent prospective cohort study indicated that the clinical alert system based on this prediction model was successfully integrated into the EHR system [91]. However, this real-time prediction model still had several limitations. First, patients whose creatinine levels were ≥4 were excluded during the development of this prediction model, but the risk and incidence of AKI and dialysis requirements are especially high in this population. Second, this prediction model did not include urine output, one of the most sensitive markers of AKI, and thus, could delay diagnosis in patients who already had oliguria but had increased serum creatinine levels. Third, more than 30 covariates were included in this prediction model; some of these covariates are infrequently checked laboratory data, such as bicarbonate and chloride levels. Moreover, as mentioned in this report, only the model containing time-updated laboratory values had similar performance in predicting AKI, sustained AKI, dialysis, and death. Unless all of these items are regularly checked in the ICU, it is difficult to evaluate AKI risk in a timely manner. Another study that performed external validation was published by Churpek et al., the data of 48,463 admissions were included in training and internal validation datasets, and the data of 447,508 admissions were used for external validation. The AUROC for predicting development AKI within 48 h was 0.72 for the internal validation cohort and the ARUROC of the two external validation cohorts were 0.67, 0.69, whereas the AUROC for predicting the receipt of renal replacement therapy within 48 h was 0.95. However, this study had a similar limitation to that of the study by Simonov et al.; the study excluded patients with serum creatinine concentration over 3.0 mg/dL on admission [87]. Higher creatinine levels and chronic kidney disease are known risk factors for AKI. It is unfortunate that the only two studies with external validation coincidentally excluded the high-risk population from the beginning.

In addition to the lack of external validation, most of the enrolled studies only predicted AKI risk at a single time point and could not provide continual predictions. Given that patients’ clinical conditions change from time to time, using laboratory, medication, and vital sign data at a single time point to perform single-point AKI risk prediction may not reflect the real-time changes of patients. One study investigated continuous risk prediction by using novel neural network algorithms. Such algorithms can process time-series data to produce time-dependent forecasts rather than forecasts that depend on summary data, as is the case in traditional methods. Tomašev et al. used the recurrent neural network to demonstrate a deep-learning approach for the continuous prediction of AKI; the approach was based on recent work on modeling adverse events from EHRs. That study was based on data provided by the United States Department of Veterans Affairs; the data were the data of 703,782 adult patients across 1243 health care facilities in the United States. By analyzing 6-hourly EHR data during hospitalization, the model predicted 55.8% of all inpatient episodes of AKI and 90.2% of all AKIs that required subsequent dialysis. The AUROC of predicting AKI within 24-, 48-, and 72-h time windows was 0.934, 0.921 and 0.914, respectively [71]. However, the high discriminative power of this system for AKI prediction derived from a large manipulated and processed dataset; the total number of independent entries in the dataset was approximately 6 billion according to the authors, which means that data cleaning and processing were difficult and had been executed by experts in data science. External validation of this successful result may be difficult due to the differing EHR systems, clinical pathways, treatments, and examination frequencies. Therefore, it may be crucial to establish an AI-assisted prediction model on the basis of a hospital’s unique clinical practices. Although real-time prediction was not performed, another study attempted to use time-series variables to improve risk prediction. Before this investigation, most postoperative AKI prediction models were based on preoperative variables. Adhikari et al. published MySurgeryRisk, a machine-learning algorithm that uses random forests to predict the postoperative AKI risk within the 3 and 7 days after surgery and the overall AKI risk. The data of 2911 patients who underwent surgery were internally validated. By combining intraoperative physiological time-series covariates with preoperative variables, machine-learning prediction models achieved an AUROC of 0.86 for predicting 7-day postoperative AKI outcomes, and AUROC was 0.84 when only the preoperative covariates of the same cohort were used. That study confirmed that postoperative AKI prediction had higher sensitivity and specificity when machine learning was applied for the dynamic incorporation of intraoperative data [72].

Most of the enrolled studies used independent cohorts; it is challenging to evaluate whether machine learning truly improved AKI risk prediction compared with the original statistics. Under this consideration, Huang et al. used the same cohort and candidate variables that were used to develop the Cath/PCI Registry AKI model as well as the data from the American College of Cardiology National Cardiovascular Data Registry collected in 1694 hospitals. That retrospective study analyzed 947,091 patients receiving PCI and concluded that the risk prediction model containing 13 variables (age, prior heart failure, cardiogenic shock within 24 h, cardiac arrest within 24 h, diabetes mellitus, coronary artery disease, heart failure within 2 weeks, preprocedure GFR and hemoglobulin, admission source, body mass index, elective or emergency PCI, and preprocedure left ventricular ejection fraction), which was validated using the generalized additive model, performed adequately, with an AUROC of 0.752 (95% CI, 0.749–0.754) and performed more highly than the original Cath/PCI Registry AKI model (AUROC, 0.711; 95% CI, 0.708–0.714). This machine-learning model also had a significantly wider predictive range than the Cath/PCI Registry AKI model did (25.3% vs. 21.6%, *p* < 0.001) and was more accurate than that model in stratifying patient risk for AKI [67].

Although machine-learning algorithms may not have matured yet and still have several limitations, they have already shown impressive performance and sensitivity in the early detection of AKI, giving clinicians useful information regarding further adverse events and long-term prognosis. By reviewing studies focused on the application of machine learning to AKI prediction, we showed that machine-learning algorithms have had a high performance for AKI prediction not only in inpatients but also in the surgical population. To date, whether the use of machine-learning algorithms for the earlier prediction of AKI risk can truly improve the prognosis of AKI remains questionable, but its ability on predicting AKI is recognized.

## 5. Conclusions

AKI is the most common and adverse potential complication of hospitalization, and it has a considerable negative impact on short-term and long-term patient outcomes. Although current guidelines use serum creatinine level and urine output rate for defining AKI and as the staging criteria of AKI, these markers are not sensitive or specific for AKI. With the advances in techniques, digitization of MISs and EHRs can provide more and timing information from patients’ underlying disease to real-time vital sign variability which increases the performance and sensitivity of machine-learning algorithms. Current studies reported that the AUROC of machine-learning algorithms on AKI prediction can be over 0.80. However, most of the studies were retrospective analyses and lacked external validation which implicated the results of the proposed models cannot be generalized outside the experimental population, and the variability of EHRs across hospitals may limit the widespread use of these prediction models. Besides, even though the MISs and EHRs provide continuous clinical records of patients but only one study performed continual risk prediction by using the recurrent neural network with a deep-learning approach, and only one study used time-series covariates to improve risk discrimination demonstrating that the use of machine learning to address large datasets is not popularized and continuous prediction of AKI via machine-learning algorithms still needs to be improved. Considering that the influencing factors, clinical and laboratory parameters might change over the hospitalization, the longitudinal evaluation to predict AKI continuously might be the next challenge of application of machine learning on AKI prediction. When the machine learning algorithms can provide real-time informatics of AKI prediction by dealing with complex databased of EHR, it might be worthwhile to look forward to the combination of machine-learning algorithms and e-alert systems. At that time, by using these machine-learning algorithms but not only serum creatinine level, e-alert systems will have a chance to provide more accurate and earlier alarm of AKI which might improve the prognosis of AKI after combining with the care bundle.

## Figures and Tables

**Figure 1 healthcare-09-01662-f001:**
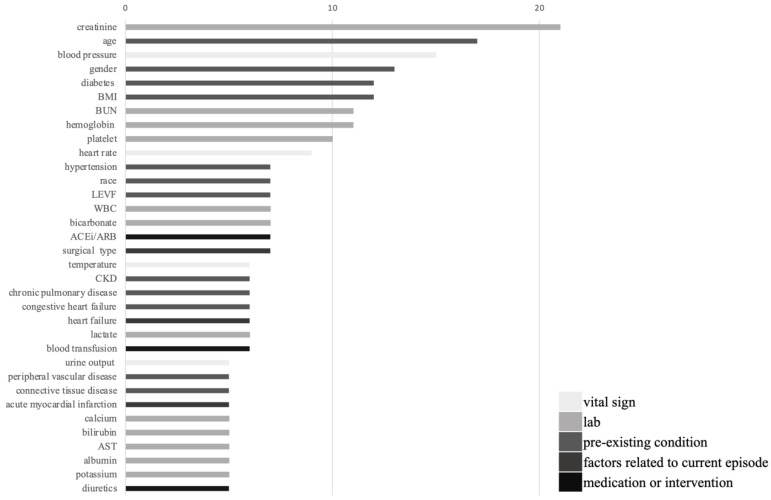
Covariates are most commonly used in machine-learning prediction models in the enrolled studies. The covariates are grouped by type. ACEi: angiotensin converting enzyme inhibitor; ARB: angiotensin receptor blocker; AST: aspartate aminotransferase; BMI: body mass index; BUN: blood urea nitrogen; CKD: chronic kidney disease; LVEF: left ventricle ejection fraction; WBC: white blood cell count.

**Figure 2 healthcare-09-01662-f002:**
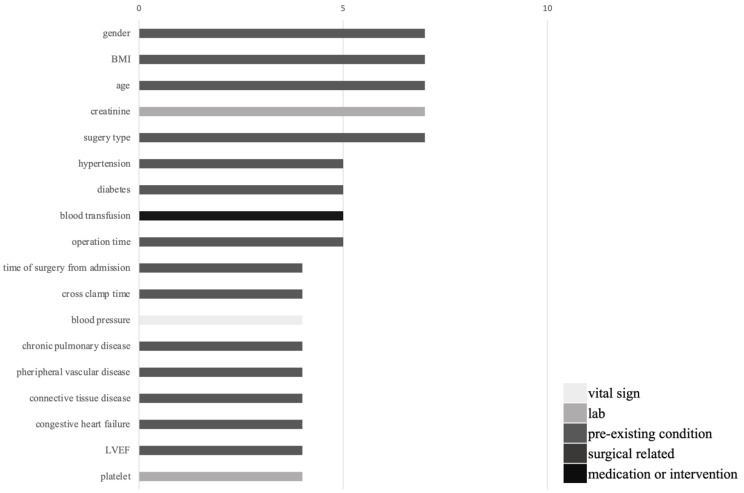
Covariates are most commonly used in machine-learning prediction models in enrolled surgical studies. The covariates are grouped by type. BMI: body mass index; LVEF: left ventricle ejection fraction.

**Table 1 healthcare-09-01662-t001:** Summary of machine-learning studies on acute kidney injury (AKI) prediction.

Scheme	Year	Design	Population	AKI Definition	Timing of AKI	AKI Incidence (%)	Patient Number	External Validation	Continuous Prediction
Kate et al. [60]	2016	retrospective	medical and surgical	AKIN	during hospitalization	8.9%	25,521	no	no
Thottakkara et al. [61]	2016	retrospective	surgical	KDIGO	post operation	36.0%	50,318	no	no
Davis et al. [62]	2017	retrospective	medical and surgical	KDIGO	during hospitalization	6.8%	2003	no	no
Cheng et al. [63]	2018	retrospective	medical and surgical	KDIGO	during hospitalization	9.0%	60,534	no	no
Ibrahim et al. [64]	2018	prospective	PCI	KDIGO	pre and post intervention	4.8%	889	no	no
Koola et al. [65]	2018	retrospective	medical and surgical	KDIGO	during hospitalization	NR (41.6% HRS)	504	no	no
Koyner et al. [66]	2018	retrospective	medical and surgical	KDIGO	24 h post admission	14.4%	121,158	no	no
Huang et al. [67]	2018	retrospective	PCI	KDIGO	during hospitalization	7.4%	947,091	no	no
Lin et al. [68]	2019	retrospective	ICU	KDIGO	during hospitalization	14%	19,044	no	no
Simonov et al. [69]	2019	retrospective	medical and surgical	KDIGO	24 h post admission	11.4–19.1%	169,859	yes	no
Huang et al. [70]	2019	retrospective	PCI	AKIN	pre and post intervention	6.4%	2,076,694	no	no
Tomašev et al. [71]	2019	retrospective	medical and surgical	KDIGO	during hospitalization	13.4%	703,782	no	yes
Adhikari et al. [72]	2019	retrospective	surgical	KDIGO	post operation	46.0%	2901	no	no
Flechet et al. [73]	2019	prospective	ICU	KDIGO	during hospitalization	12% †	252	no	no
Parreco et al. [74]	2019	retrospective	medical and surgical	KDIGO	during hospitalization	5.6%	151,098	no	no
Xu et al. [75]	2019	retrospective	medical and surgical	KDIGO	during hospitalization	NR	58,976	no	no
Tran et al. [76]	2019	prospective	burn	KDIGO	during hospitalization	50.0%	50	no	no
Zhang et al. [77]	2019	retrospective	ICU	KDIGO	24 h post admission	58.1%	6682	no	no
Zimmerman et al. [78]	2019	retrospective	ICU	KDIGO	72 h post admission	16.5%	46,000	no	no
Rashidi et al. [79]	2020	retrospective and prospective	burn and trauma	KDIGO	1st week post ICU admission	50.0%	101	no	no
Zhou et al. [80]	2020	retrospective	TAAAR	NR	post operation	12.7%	212	no	no
Martinez et al. [81]	2020	retrospective	medical and surgical	KDIGO	emergency department	7.9%	59,792	no	no
Lei et al. [82]	2020	retrospective	TAAR	KDIGO	post operation	72.6%	897	no	no
Lei et al. [83]	2020	retrospective	hepatectomy	KDIGO	post operation	6.6%	1173	no	no
Qu et al. [84]	2020	retrospective	acute pancreatitis	KDIGO	during hospitalization	24.0%	334	no	no
Tseng et al. [85]	2020	retrospective	Cardiac surgery	KDIGO	post operation	24.3%	671	no	no
Sun et al. [86]	2020	retrospective	PCI	KDIGO	during hospitalization	15.1%	1495	no	no
Churpek et al. [87]	2020	retrospective	medical and surgical	KDIGO	during hospitalization	14.3%	495,971	yes	no
Hsu et al. [88]	2020	retrospective	medical and surgical	KDIGO	Community acquired AKI	8.4%	234,867	no	no
Penny-Dimri et al. [89]	2020	retrospective	Cardiac surgery	Other *	post operation	6.5%	97,964	no	no
Li et al. [90]	2020	retrospective	Cardiac surgery	KDIGO	post operation	37.5%	5533	no	no

* The AKI definition in this study was as follows: (1) new postoperative and in-hospital serum creatinine level > 200 mmol/L AND a doubling or greater increase in creatinine over the baseline preoperative value AND the patient did not require preoperative renal replacement therapy; and (2) a new inhospital requirement for renal replacement therapy. † Only reported the percentage of AKI stage 2 and stage 3. AKI: acute kidney injury; ICU: intensive care unit; PCI: percutaneous coronary intervention; TAAR: total aortic arch replacement; TAAAR: thoracoabdominal aortic aneurysm repair.

**Table 2 healthcare-09-01662-t002:** Summary of data processing and performance of machine-learning algorithm in enrolled studies.

Study	Feature Selection Algorithm	Feature Selection Method	Data Splitting	Machine Learning Algorithm	AUROC
Kate et al. [60]	NR	NR	ten-fold cross-validation	naïve Bayes	0.654
				SVM	0.621
				decision trees	0.639
				logistic regression	0.660
Thottakkara et al. [61]	LASSO	embedded method	training data (70%); validation (30%)	naïve Bayes	0.819
				generalized additive model	0.858
				logistic regression	0.853
				support vector machine	0.857
Davis et al. [62]	according to clinical experience or previous report	NR	five-fold cross-validation	random forest	0.73
				neural network	0.72
				naïve Bayes	0.69
				logistic regression	0.78
Cheng et al. [63]	according to clinical experience or previous report	NR	ten-fold cross-validation	random forest	0.765
				AdaBoostM1	0.751
				logistic regression	0.763
Ibrahim et al. [64]	LASSO	embedded method	Monte Carlo cross-validation	logistic regression	0.79
Koola et al. [65]	LASSO	embedded method	five-fold cross-validation	logistic regression	0.93
				naïve Bayes;	0.73
				support vector machines;	0.90
				random forest;	0.91
				gradient boosting	0.88
Koyner et al. [66]	tree-based method	embedded method	ten-fold cross-validation	gradient boosting	0.9
Huang et al. [67]	XGBoost and LASSO	embedded method	training data (70%); validation (30%)	gradient boost;	0.728
				logistic regression	0.717
Lin et al. [68]	according to clinical experience or previous report	NR	five-fold cross-validation	SVM	0.86
Simonov et al. [69]	according to clinical experience or previous report	NR	training data (67%); validation (33%)	discrete-time logistic regression	0.74
Huang et al. [70]	stepwise backward selection, LASSO, premutation-based selection	embedded method	training (50%); validation (50%)	generalized additive model	0.777
Tomašev et al. [71]	L1 regularization	embedded method	training (80%); validation (5%); calibration (5%); test (10%)	recurrent neural network	0.934
Adhikari et al. [72]	F-test	filter method	five-fold cross-validation	random forest	0.86
Flechet et al. [73]	according to clinical experience or previous report	NR	NR	random forest	0.78
Parreco et al. [74]	NR	NR	NR	gradient boosting;	0.834
				logistic regression;	0.827
				deep learning	0.817
Xu et al. [75]	gradient boosting	embedded method	five-fold cross-validation	gradient boosting	0.749
Tran et al. [76]	NR	NR	Scikit-learn cross validation	k-nearest neighbor	0.92
Zhang et al. [77]	XGBoost	embedded method	bootstrap validation	gradient boosting	0.86
Zimmerman et al. [78]	logistic regression	embedded method	five-fold cross-validation	logistic regression	0.783
				random forest	0.779
				neural network	0.796
Rashidi et al. [79]	according to clinical experience or previous report	NR	Scikit-learn cross validation	recurrent neural network	0.92
Zhou et al. [80]	NR	NR	five-fold cross-validation	logistic regression	0.73
				linear kernel SVM	0.84
				Gaussian kernel SVM	0.77
				random forest	0.89
Martinez et al. [81]	LASSO	embedded method	ten-fold cross-validation	random forest	not provided
Lei et al. [82]	NR	NR	training data (70%); validation (30%)	Gradient boosting	0.8
Lei et al. [82]	NR	NR	training data (70%); validation (30%)	Gradient boosting	0.772
				Light gradient boosted machine	0.725
				random forest	0.662
				DecisionTree	0.628
Qu et al. [84]	NR	NR	ten-fold cross-validation	random forest	0.821
				classification and regression tree	0.8033
				logistic regression	0.8728
				extreme gradient boosting	0.9193
Tseng et al. [85]	tree-based method	embedded method	five-fold cross-validation	random forest	0.839
				random forest with extreme gradient boosting	0.843
Sun et al. [86]	Boruta algorithm	wrapper method	ten-fold cross-validation	random forest	0.82
				logistic regression;	0.69
Churpek et al. [87]	gradient boosting	embedded method	ten-fold cross-validation	gradient boosted machine	0.72
Hsu et al. [88]	XGBoost and LASSO	embedded method	five-fold cross-validation	logistic regression;	0.767
Penny-Dimri et al. [89]	tree-based method	embedded method	five-fold cross-validation	logistic regression;	0.77
				gradient boosted machine	0.78
				neural networks	0.77
Li et al. [90]	LASSO	embedded method	ten-fold cross-validation	Bayesian networks	0.736

AUROC: area under the receiver operating characteristic curve; LASSO: least absolute shrinkage and selection operator; NR: not reported; SAPS: simplified acute physiology score; SVM: support vector machine; XGB: eXtreme Gradient Boostin.

## Data Availability

Not applicable.

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
