# Peer review of "Does Artificial Intelligence Make Clinical Decision Better? A Review of Artificial Intelligence and Machine Learning in Acute Kidney Injury Prediction"

_healthcare, 2021, doi:10.3390/healthcare9121662_

Round 1
Reviewer 1 Report
The authors present a fairly thorough review of the literature related to the diagnosis of AKI. However, they did not avoid several significant errors that need to be corrected before possible publication.
1. The classic IMRAD structure in the article is missing. The authors should, for example, refer to other review papers in the same journal e.g.
https://www.mdpi.com/2227-9032/9/10/1381/htm
2. The conclusion contains 3 sentences of which 2 are repetitions from the abstract!
The last sentence of the conclusion should, in my opinion, be expanded, as on its own in the conclusion is an obvious statement (more data - chance for better prediction):
„With advances in MISs and EHRs, machine-learning algorithms have become increasingly attractive means of AKI prediction; the performance and sensitivity of prediction models obtained using these methods indicate that when combined with care bundles, they can be used to treat and prevent AKI early”
The paragraph between lines 360-380 seems to be kind of discussion and conclusions mixed with results.
It would be useful to provide in the conclusions some summary quantitative estimates from the results by how much there is a chance to improve the diagnosis of AKI using additional parameters.
The abstract should also include a more quantitative description of the results.
3. The results presented are a tabulation of data from various publications. However valuable, they do not contain information relevant to applications and machine learning. E.g. covariates are presented only in terms of their occurrence in publications, but there is no information or discussion on what basis they were selected (what methods were used in the selection e.g. ICA, PCA or derivatives, or simply because they were readily available?). The lack of discussion on this topic may lead to erroneous conclusions.
4. Minor editing errors e.g "." period in line 46 before quoting. All figures and its captions mixed with text (line: 261,267, 382). Colors used in fig. 1 and 2 are not well "distinguishable". Instead of using shades of blue, it is probably better to use shades of grey. Figure 3 as a general scheme should be considered as introductory illustration otherwise it is unnecessary.
Author Response
Reviewer #1:
- The classic IMRAD structure in the article is missing. The authors should, for example, refer to other review papers in the same journal e.g.
https://www.mdpi.com/2227-9032/9/10/1381/htm
Reply:
Thank you for your kindly comment. We had modified our manuscript to the IMRAD structure as your suggestion.
- The conclusion contains 3 sentences of which 2 are repetitions from the abstract! The last sentence of the conclusion should, in my opinion, be expanded, as on its own in the conclusion is an obvious statement (more data - chance for better prediction): „With advances in MISs and EHRs, machine-learning algorithms have become increasingly attractive means of AKI prediction; the performance and sensitivity of prediction models obtained using these methods indicate that when combined with care bundles, they can be used to treat and prevent AKI early”
The paragraph between lines 360-380 seems to be kind of discussion and conclusions mixed with results.
It would be useful to provide in the conclusions some summary quantitative estimates from the results by how much there is a chance to improve the diagnosis of AKI using additional parameters.
The abstract should also include a more quantitative description of the results.
Reply:
Thank you for your valuable comment. We had modified the sentences in the abstract, results and conclusion sessions following your suggestion. The modified sentence was in page 1, line 16-19; page 15, line 408-411 and page 16, line 417-438 in the revised manuscript.
- The results presented are a tabulation of data from various publications. However valuable, they do not contain information relevant to applications and machine learning. E.g. covariates are presented only in terms of their occurrence in publications, but there is no information or discussion on what basis they were selected (what methods were used in the selection e.g. ICA, PCA or derivatives, or simply because they were readily available?). The lack of discussion on this topic may lead to erroneous conclusions.
Reply:
Thank you for your valuable comment. According to your suggestion, we had added Table 2 which summarized the feature selection, data splitting and machine learning algorithm type in enrolled studies to provide more information and added a short paragraph to discuss the covariates choice in these studies. However, only three studies mentioned that they use PCA for feature extraction, other studies did not report the method they used in feature extraction. The added Table was in page10-12 in the revised manuscript.
- Minor editing errors e.g "." period in line 46 before quoting. All figures and its captions mixed with text (line: 261,267, 382). Colors used in fig. 1 and 2 are not well "distinguishable". Instead of using shades of blue, it is probably better to use shades of grey. Figure 3 as a general scheme should be considered as introductory illustration otherwise it is unnecessary.
Reply:
Thank you for your kindly comment. We had corrected the editing errors you mentioned and changed fig 1 and 2 colors to shades of grey. For the figures’ placed, the Healthcare author guideline request that all figures should be cited in the main text and placed near to the first time they are cited, so we keep their position in order to follow the journal instruction. We also removed fig 3 as your suggestion.

Reviewer 2 Report
This review is devoted to a topical topic of clinical medicine - predicting the development of acute kidney injury. The existing criteria for the development of acute kidney injury do not allow them to be used to predict the development of this complication. The authors of the review emphasize that advances in computing technology have led to the use of machine learning and artificial intelligence in AKI prediction. In the review, the authors consider studies devoted to this problem, including those with external validation of predictive models, and also consider their limitations. I believe that the article can be useful for clinicians and researchers in this area.
Author Response
Reviewer:
This review is devoted to a topical topic of clinical medicine - predicting the development of acute kidney injury. The existing criteria for the development of acute kidney injury do not allow them to be used to predict the development of this complication. The authors of the review emphasize that advances in computing technology have led to the use of machine learning and artificial intelligence in AKI prediction. In the review, the authors consider studies devoted to this problem, including those with external validation of predictive models, and also consider their limitations. I believe that the article can be useful for clinicians and researchers in this area.
Reply:
Thank you for your kindly comment. We hope that our article can provide more useful information for clinicians and researchers putting their effort to prevent and early treat AKI.

Reviewer 3 Report
The review made by Tao Han Lee and collegues is well written, complete and very informative. It is quite similar to the revue done by Joana Gameiro et al in march 2020 excepted they have collected some more studies (31 versus 19). Few points need to be re-enforced or discussed:
- To understand the need to introduce AI in the comprehension of risk factors of AKI it would be interesting to highlight the diversity of the different studies done and the different population in paragraph 3 leading to the need of new strategies and neuronal networks to hand up all diversities and parameters.
- Paragraph 4, the authors discuss the development of electronic alerts. Could the authors indicate some example and type of alerts used for the diagnostic of AKI or for it management and their benefices for the management of AKI.
- AKI is a very interesting situation in which all the markers or parameters are not necessary all present at one time but can be acquired during hospitalization because of the worsening of the disease, the toxicity of molecules used or the adapted therapeutical strategies developed by physicians. The authors have suggested the need to integrate a longitudinal evaluation. This is an important point that should be re-enforced in the conclusion as well as the integration of neuronal networks and the necessity to also integrate the therapeutics used for the management of AKI
Author Response
Reviewer:
The review made by Tao Han Lee and collegues is well written, complete and very informative. It is quite similar to the revue done by Joana Gameiro et al in march 2020 excepted they have collected some more studies (31 versus 19). Few points need to be re-enforced or discussed:
- To understand the need to introduce AI in the comprehension of risk factors of AKI it would be interesting to highlight the diversity of the different studies done and the different population in paragraph 3 leading to the need of new strategies and neuronal networks to hand up all diversities and parameters.
Reply:
Thank you for your kindly comment. We modified this paragraph and added the studies investigating the variety of AKI incidence between different races, geographic and even economic populations to emphasize that race and epidemiology factors had its role in AKI prediction. The modified sentence was in page 4, line 169-183, in the revised manuscript as follows:
“Besides the comorbidities and acute illness condition, race and epidemiology factors also showed its impact on AKI incidence according to previous studies. Mathioudakis and his colleagues had reported that blacks had a 50% higher age- and sex-adjusted odds of AKI compared to whites (odds ratio: 1.51; 95% CI 1.37–1.66) based on the national databases of U.S. This association between black race and increased risk of AKI persisted after additional adjustment for multiple AKI-related risk factors. In 2013, a meta-analysis focused on AKI incidence worldwide reported that the pooled rate of AKI according to KDIGO criteria showed a difference around the world. According to geographic regions of the world and patterns of country economies and latitude, the pooled rate of AKI appeared higher in South versus North America (29.6% versus 24.5%), Southern versus Northern Europe (31.5% versus 14.7%), and South versus Western or Eastern Asia (23.7% versus 16.7% versus 14.7%). The pooled rate of AKI appeared higher in studies from countries located south versus north of the equator (27.0% versus 22.6%), in addition, this study also revealed that the AKI incidence was high in countries that spent >10% versus ≦5% GDP on total health expenditure (25.2% versus 14.5%).”
- Paragraph 4, the authors discuss the development of electronic alerts. Could the authors indicate some example and type of alerts used for the diagnostic of AKI or for it management and their benefices for the management of AKI.
Reply:
Thank you for your valuable comment. We had added some detailed information and results about previous studies focused on e-alert system in this paragraph to provide more information about e-alert systems application. The modified sentence was in page 5, line 216-230, in the revised manuscript as follows:
“Park and his colleagues had investigated an AKI alert system with automated nephrologist consultation in which clinicians could generate automated consultations to the nephrology division while patient’s serum creatinine concentration elevation of at least 1.5-fold or 0.3 mg/dL from baseline. This study reported that the early consultation with a nephrologist was greater (adjusted OR, 6.13; 95% CI, 4.80- 7.82) and odds of a severe AKI event was reduced (adjusted OR, 0.75; 95% CI, 0.64-0.89) after introducing the e-alert system. But mortality was not affected (adjusted HR, 1.07; 95% CI, 0.68-1.68). Another study used an e-alert system in ICU patients, clinician received a “pop-up” message while e-alert system screening the serum creatinine data and detected possible AKI events followed the KDIGO criteria definition. Although the sensitivity, specificity, Youden Index and accuracy of the AKI e-alert system were 99.8, 97.7, 97.5 and 98.1%, respectively, in this study, and the prevalence of diagnosis AKI and the prevalence of nephrology consultation in the e-alert group was higher than that in the non-e-alert group. There was no significant difference in the prevalence of dialysis, rehabilitation of renal function or death in the two groups.”
- AKI is a very interesting situation in which all the markers or parameters are not necessary all present at one time but can be acquired during hospitalization because of the worsening of the disease, the toxicity of molecules used or the adapted therapeutical strategies developed by physicians. The authors have suggested the need to integrate a longitudinal evaluation. This is an important point that should be re-enforced in the conclusion as well as the integration of neuronal networks and the necessity to also integrate the therapeutics used for the management of AKI
Reply:
Thank you for your valuable comment. We had modified several sentences in the conclusions paragraph and emphasize the important of longitudinal evaluation on AKI prediction. The modified sentence was in page 16, line 430-432, in the revised manuscript as follows:
“Considering that the influencing factors, clinical and laboratory parameters might change over the hospitalization, the longitudinal evaluation to predict AKI continuously might be the next challenge of application of machine learning on AKI prediction.”
